

# Method selection affects the estimates of residency and site fidelity in bottlenose dolphins: testing sensitivity and performance of different methods using mark-resight data

Israel Huesca-Domínguez[1,2], Eduardo Morteo[1,2], Luis Gerardo Abarca-Arenas[1], Brian C. Balmer[3], Tara M. Cox[4], Christian A. Delfín-Alfonso[1,2,5] and Isabel C. Hernández-Candelario[1,2]

[1] Instituto de Investigaciones Biológicas, Universidad Veracruzana, Xalapa, Veracruz, Mexico

[2] Laboratorio de Mamíferos Marinos (LabMMar), Instituto de Investigaciones Biológicas, Universidad Veracruzana, Xalapa, Veracruz, Mexico

[3] Dolphin Relief and Research, Clancy, MT, United States of America

[4] Department of Marine and Environmental Sciences, Savannah State University, Savannah, GA, United States of America

[5] Laboratorio de Vertebrados, Instituto de Investigaciones Biológicas, Universidad Veracruzana, Xalapa, Veracruz, Mexico

Corresponding author
Eduardo Morteo, emorteo@uv.mx

## ABSTRACT

Residency (R) and site fidelity (SF) are important parameters in population ecology, yet their quantification poses challenges in marine mammals. Based on a previous review, this study used simulated and empirical mark-resight data to assess the variations and performance of the most used R ($n = 8$) and SF ($n = 11$) indices in peer-reviewed literature under different scenarios. We applied the Jolly-Seber model to simulate thousands of bottlenose dolphin populations varying resighting ($p$) and survival ($Phi$) probabilities, and performed calibration, sensitivity, and validation analyses. Our results underscore the effects of $p$ and $Phi$ on individual categorization within the diverse simulated conditions, representing the often-overlooked heterogeneity in residency classification for *Tursiops* populations. All SF indices showed similar and consistent performance ($>0.70$ Gower's distance) across the simulated scenarios, even when compared to field study data from wild dolphin populations (*i.e.,* Savannah, USA, and Alvarado, Mexico); thus, SF should be a critical parameter for interstudy comparisons. Conversely, R indices were remarkably different based on their definitions and classification criteria. The different thresholds among definitions largely biased the proportion of residents and transient individuals (or occasional visitors) even leading to counterintuitive outcomes. This emphasizes the importance of considering trade-offs in R index selection aligned with project goals, specific sampling efforts, and population dynamics. For instance, the simplified binomial categorization of R defined by Conway (2017) (https://digitalcommons.coastal.edu/etd/10/) easier to interpret but R indices incorporating temporal components (*e.g.,* monthly, seasonal, and annual) outperformed ($>0.70$ Gower's distance) other R indices lacking such criteria. This allowed for a more detailed representation of the temporal structure of the population, and higher consistency and accuracy while classifying individuals. Also,
although the residency categories proposed by Möller, Allen & Harcourt (2002) (DOI 10.1071/AM02011) did not perform as well, these seemed to fit better when dealing with data gaps across spatial and temporal scales. Our results contribute to the ongoing discussion on methodological implications for the interpretation of ecological patterns, facilitating a nuanced understanding of population dynamics, aiding scientists, and conservation agencies in making informed decisions for bottlenose dolphin populations worldwide.

# INTRODUCTION

Recurrent presence and long-term permanence of individuals are crucial data for ecologists and resource managers to better understand distribution patterns, habitat use, and anthropogenic threats that animal populations may be exposed to (*Azzellino et al., 2017*; *Vermeulen et al., 2017*; *La Manna et al., 2023*). In general, individuals with long-term permanence are likely better adapted to fine-scale habitat characteristics; however, they may also be more vulnerable to acute stochastic disturbances that are becoming more frequent due to recent environmental and climatic changes, including those of anthropogenic nature (*Morris & Beer, 2003*; *Poeta et al., 2017*; *Sousa et al., 2019*; *Cloyed et al., 2021*; *Lettrich et al., 2023*).

For decades, capture-recapture methods using natural markings on the dorsal fin and/or body of cetaceans species have been a standardized methodology to better understand population dynamics, including abundance, social interactions, and movement patterns (*Urian et al., 2014*; *Hammond et al., 2021*), but also long-term permanence or frequent occurrence, which are commonly referred to in the literature as residency (R) and site fidelity (SF) (*e.g.*, *Belda, Barba & Monrós, 2007*; *Balmer et al., 2008*; *Boucek et al., 2019*; *Griffin et al., 2021*). In recent years, assessing R and SF have provided more detailed information about how these parameters may influence population susceptibility to various stressors, including biotoxins, contaminants, disease, fisheries interactions, and noise (*e.g.*, *Balmer et al., 2008*; *Rosel et al., 2011*; *Bolaños Jiménez et al., 2021*; *Trabue, Rekdahl & Rosenbaum, 2024*). However, the details on how to measure R and SF are largely subjective and somewhat limited, especially in marine mammals such as bottlenose dolphin populations (*Ballance, 1990*; *Morteo, Rocha-Olivares & Morteo, 2012*; *Tschopp et al., 2018*; *Huesca-Domínguez et al., 2024*). The lack of consensus in the definitions, sampling methodology, and standardization of metrics used to collect identification data on individuals with mark-resight techniques presents a great challenge (*Morteo, Rocha-Olivares & Morteo, 2012*; *Urian et al., 2014*; *Tschopp et al., 2018*; *Szott, Brightwell & Gibson, 2022*).

The different methods of estimating R and SF in bottlenose dolphins are particularly difficult to adopt when sampling effort is low, considering that a large proportion of

individuals may leave the study area within the survey period (*Silva et al., 2008*; *Rosel et al., 2011*; *Morteo, Rocha-Olivares & Morteo, 2012*; *Conway, 2017*); furthermore, this task is even harder when populations have extended ranging patterns (*e.g., Balmer et al., 2014*; *Balmer et al., 2018*; *Speakman et al., 2023*). Therefore, extensive sampling effort and availability of historically reliable sighting data for individuals in a particular area are of great importance, as they influence the accuracy of estimates in population parameters, including R and SF (*Bejder et al., 2006*; *Speakman et al., 2010*; *Urian et al., 2014*; *Conway, 2017*; *Nykänen et al., 2020*; *Bolaños Jiménez et al., 2021*).

Residency is usually defined as the timeframe an animal spends in a specific area (*Wells & Scott, 1990*); whereas site fidelity is commonly defined as the tendency of an animal to return to a previously occupied area (*Mayr, 1963*; *Switzer, 1993*). Since the early 1990s, different metrics to evaluate R and SF have been proposed according to the sampling needs and research goals of a project using mark-resight methods in wild bottlenose dolphins (*Ballance, 1990*; *Rossi-Santos, Wedekin & Monteiro-Filho, 2007*; *Huesca-Domínguez et al., 2024*). These indices have different qualitative or quantitative units such as frequencies, categories, and proportions, as well as models that can also incorporate varying temporal scales (*e.g.*, *Zolman, 2002*; *Rosel et al., 2011*; *Chabanne et al., 2012*; *Tschopp et al., 2018*). However, to date, there is little consistency in the optimal method to estimate these parameters, and biases are known to occur due to differences in sampling frequency, study area size, sampling duration, and individual sighting probabilities (*Morteo, Rocha-Olivares & Morteo, 2012*; *Tschopp et al., 2018*; *Caruso et al., 2024*). Thus, a comprehensive analysis is necessary, including an evaluation of R and SF definitions through an experimental design that allows standardization of existing metrics with different values of effort across spatial and temporal scales (*Conway, 2017*; *Huesca-Domínguez et al., 2024*).

The goal of this study was to quantitatively assess variations in the outcomes of the most common indices of R and SF for bottlenose dolphin populations in peer-reviewed literature under different simulated scenarios using mark-resight data. In addition, we calibrated the different metrics using *a priori* data for population parameters on wild populations of the genus *Tursiops* (based on specialized literature). Finally, we used these simulations to validate and assess the performance of these indices under the different scenarios.

## METHODS

### Metrics

R and SF assessments were first separated based on their actual use as expressed by the authors in peer-reviewed literature. However, given the wide variety of criteria among studies (see *Huesca-Domínguez et al., 2024*), we started with the simplest concept and thus considered SF indices as those using sighting frequencies regardless of temporal scale and/or sampling periodicity to classify individuals (Table 1). The latter resulted in 11 indices to assess SF, based on different types of variables (*i.e.,* categorical in nominal or ordinal scale, or numerical).

Conversely, definitions in which individuals were directly classified using different measures of occurrence within a given timeframe (monthly, temporary, seasonal, and/or

**Table 1** **Description of the parameters to assess site fidelity (SF) indices in dolphins.** This include authors, description and type (categorical, ordinal, nominal, numerical, *etc.*)

| Author | Description | Type of variable |
|---|---|---|
| *Balmer et al. (2008)* | Classification of the number of sightings according to the underlying density distribution of the data. | Categorical, Ordinal |
| *Chabanne et al. (2012)* | Monthly sighting rate (MSR). | Numerical (both variables) |
| | Seasonal sighting rate (SSR). | |
| *Díaz-López (2012)* | Arbitrary categories from sighting rates are based on the temporal occurrence rates. | Categorical, Nominal |
| *Möller, Allen & Harcourt (2002)* | Arbitrary categories from sighting rates based on the temporal occurrence | Categorical, Nominal |
| *Ballance (1990)* | Occurrence: number of sighting records. | Numerical |
| *Parra, Corkeron & Marsh (2006)* | Annual sighting rate. | Numerical |
| *Ballance (1990)* | Periodicity: mean time (days) between consecutive resightings. | Numerical |
| *Morteo, Rocha-Olivares & Morteo (2012)* | Periodicity: mean time (days) between consecutive resightings. | Numerical |
| *Ballance (1990)* | Permanence: time (days) between the first and last sighting. | Numerical |
| *Quintana-Rizzo & Wells (2001)* | Arbitrary categories from occurrence. | Multi-categorical (fuzzy) |
| *Culloch (2004)* | Arbitrary categories from occurrence. | Multi-categorical (fuzzy) |
| *Tschopp et al. (2018)* | Combined index from Ballance's parameters of occurrence and permanence. | Numerical |

**Table 2** **Description of the parameters to assess residency (R) indices in dolphins.** This includes authors, descriptions and type (categorical, ordinal, nominal, numerical, *etc.*)

| Author | Description | Type of variable |
|---|---|---|
| *Ananias, Jesus & Yamamoto (2008)* | Classification of residency according to monthly sightings. | Multiple categories (fuzzy) |
| *Chabanne et al. (2012)* | Classification of residency according to monthly and seasonal sightings. | Categorical, Nominal |
| *Conway (2017)* | Classification of residency according to sighting time by season. | Multiple categories (fuzzy) |
| *Dinis et al. (2016)* | Classification of residency according to seasonal and annual sightings. | Categorical, Nominal |
| *Martin & Silva (2004)* | Classification of residency according to monthly and annual sightings. | Categorical, Nominal |
| *Möller, Allen & Harcourt (2002)* | Classification of residency according to sighting rates. | Categorical, Nominal |
| *Rosel et al. (2011)* | Classification of residency according to sighting time. | Multiple categories (fuzzy) |
| *Zolman (2002)* | Classification of residency according to seasonal sightings. | Categorical, Nominal |

annual) were defined as R indices. Thus, eight R indices corresponded to criteria provided by peer-reviewed publications that defined different classifications of individuals (Table 2). All of these contained a categorical scale with two (residents and non-residents) or three (residents, non-residents, and occasional/transient individuals) types of individuals based on their mark-resight histories.
## Description of the model

Mark-resight models to estimate population size, survival and other demographic variables (*Nichols, 1992*) are based on the idea that each member of the population has a sighting history $Y_i = (Y_{i1}, \ldots, Y_{it})$, where $Y_{it} = 1$ if individual $i$, is sighted at sampling occasion $t$, $t = 1, \ldots, T$, and otherwise $Y_{it} = 0$ (see *Morteo, Rocha-Olivares & Morteo, 2012*). One of the most common models is the Jolly-Seber (J-S) (*Schwarz, 2001*), which has been widely used in a broad range of taxa (*e.g.*, *Vargas-Fonseca et al., 2020*; *Bolaños Jiménez et al., 2021*; *Ali & Rajamani, 2023*; *Blázquez et al., 2023*), and thus developed, parametrized, extended, and implemented by different researchers across years and study areas.

We chose the algorithm developed by *Kéry & Schaub (2012)* in the R-project language (simul.js) since it generates sighting histories under the implemented assumptions of the J-S model. The model is based on parameterizations of superpopulations (*i.e.,* the number of animals that use the study area at some point during the study) with constant sighting and survival probabilities and time-dependent entry probabilities (see *Williams, Frederick & Nichols, 2011*; *Bolaños Jiménez et al., 2021*).

Following the mark-resight methods, the algorithm simulates sampling strategies based on the identification of individuals (marking) in a target population, to take subsequent samples under the same conditions and determines what fraction of these were identified on different occasions (resightings) (*Seber & Schofield, 2019*).

The probability that an individual that has not been present since the previous occasion enters the population on occasion t is $b_t (t = 1, \ldots, T)$ and is called the entry probability (*Schwarz & Arnason, 1996*). Entry could result from on-site recruitment (individuals born locally) or immigration (individuals born outside the study area entering at some point). To ensure that all individuals ($N$) enter the population at some point during the study, the sum of all entry probabilities must be equal to 1 (*Kéry & Schaub, 2012*).

The algorithm provides simulated sighting histories of the $N$ individuals, along with the number of individuals that entered the population for the first time and the actual size of the population on each occasion, according to the input parameters provided by the user.

## Calibration by simulating scenarios

Calibration was obtained by identifying the optimal values of the parameters from the model, such that its fit was maximized with respect to a set of simulated data (*Campbell, 2006*; *Trucano et al., 2006*). In this sense, we used fixed and variable values for parameters to build sighting histories of individuals in simulated wild dolphin populations (see *Morteo, Rocha-Olivares & Morteo, 2012*). Input parameters for the algorithm (simul.js) were: (1) size of the superpopulation ($N$), (2) probability of entry ($b_t$), (3) resighting probability ($p$), (4) apparent survival probability ($Phi$), (5) number of marked individuals ($n$), (6) number of sampling occasions ($t_i$) and (7) sampling frequency ($m$). Since dolphin populations around the world can range from very small to over 400 individuals, we used a superpopulation size of 600 individuals (*Morteo, Rocha-Olivares & Morteo, 2012*; *Caruso et al., 2024*), which resulted in random sizes of marked populations ranging from a few individuals to over 400, according to the recapture and survival probabilities tested here. The probability of entry (or recruitment) consisted of a random value for each sighting history varying between
0.05 and 0.15; this range was used to represent stochastic variations in the population and sampling effort (*Kéry & Schaub, 2012*; *Morteo, Rocha-Olivares & Morteo, 2012*), given that the proportion of the population that is first identified is rather small, but increases over time with an asymptotic trend once sufficient time and sampling effort have occurred (*Morteo, Rocha-Olivares & Morteo, 2012*).

Resighting probability ($p$) between sampling occasions was set from 0.01 to 1.00 in increments of 0.10, to represent naturally variable scenarios, considering individuals present only once (*i.e.,* 0.01), those present on different occasions in the study period, and also animals sighted in all samples (*i.e.,* 1.00). On the other hand, survival probabilities (*Phi*) varied between 0.70 and 1.00 (with 0.05 increments) and were considered to represent the biological effect of sex and age in bottlenose dolphins due to survival heterogeneity between males and females, as well as among developmental stages (*i.e.,* newborn, calf, juvenile, and adult). Survival values reported in the literature are typically higher in females than in males, and lower for calves and juveniles, with trends showing values above 0.7 in all cases (*Currey et al., 2009*; *Ludwig et al., 2021*; *Arso Civil et al., 2019*; *Bolaños Jiménez et al., 2021*; *Barratclough et al., 2024*; *Schwacke et al., 2024*). Also, the sampling duration was set at three years, considering the mean duration obtained in the reviews by *Morteo, Rocha-Olivares & Morteo (2012)* and ensuring full-time coverage to assess residency using indicators that require at least one or two years of sampling (*Huesca-Domínguez et al., 2024*). Finally, sightings were clustered using monthly (*i.e.,* 36 occasions), seasonal (*i.e.,* 12 conventional seasons), and annual (*i.e.,* three years) periods to make all calculations.

## Sensitivity analysis

In the sensitivity analysis, the input parameters of the model were changed to determine the effect on the behavior of the output values (*Sargent, 2010*). To this end, sighting histories were originally set to 10,000 simulations following *Morteo, Rocha-Olivares & Morteo (2012)*. The sensitivity of the different definitions of R and SF within the mark-resight model were assessed by changing survival (*Phi*) and resighting ($p$) values within the given ranges (0.01–1.00 and 0.70–1.00, respectively) for each unique combination of the fixed values of the model (*i.e.,* population size, entry probability, and number of sampling occasions) to obtain the proposed sets of simulated sighting histories with the same features. Thus, in each of these simulated scenarios, we applied the most common definitions of R and SF to the same data (Tables 1 and 2). We emphasized the variables originally proposed by *Ballance (1990)*, for being among the first and most widely used in the literature (*Morteo, Rocha-Olivares & Morteo, 2012*; *Huesca-Domínguez et al., 2024*). Subsequently, we used the estimates of each index for the same simulated data, and averaged the proportion of individuals within each R and SF category to analyze the sensitivity of the estimators to variations of the implemented parameters (*i.e.,* resighting and survival); that is, if the curves of a given index under different values of $p$ and *Phi* do not change, then the index is insensitive to changes in those parameters over the given range of values. Conversely, if the curve changes drastically as the level of the parameters changes, then the estimator is sensitive to that parameter.

## Case studies

We used individual sighting histories from two long-term photographic-identification (Photo-ID) studies on wild populations of *Tursiops spp.* (Savannah, USA, and Alvarado, Mexico), to compute empirical metrics of residency and site fidelity. Based on the methodology proposed by *Morteo, Rocha-Olivares & Morteo (2012)*, we used data from three consecutive years of study for each population. Sampling efforts from both populations had different frequencies which allowed us to assess the performance of the indices under different empirical scenarios.

The first case study corresponded to data from the common bottlenose dolphin (*Tursiops truncatus*) photo-ID by the Marine Mammal Laboratory at Universidad Veracruzana (LabMMar-IIB-UV) from May 2006 to August 2010 (*Morteo, 2011*; *Morteo, Rocha-Olivares & Morteo, 2012*; *Morteo, Rocha-Olivares & Abarca-Arenas, 2014*; *Morteo, Rocha-Olivares & Abarca-Arenas, 2017*). For this research, monthly sighting histories from January 2007 to November 2009 were used, and the study area was covered on each survey, but the dataset contained months during which no sampling was conducted. Nevertheless, information was available for all seasons of the year. In this case, 206 individuals were identified in 29 months across the three years.

The second case corresponded to data from the Tamanend's bottlenose dolphin (*Tursiops erebennus*) photo-ID in estuaries from the southern Savannah River to the northern Ossabaw Sound, in Georgia, USA, from April 2009 to July 2017 (*Perrtree, Kovacs & Cox, 2014*; *Kovacs, Perrtree & Cox, 2017*). The sampling period covered January 2014 to December 2016, with monthly consecutive surveys. Efforts were made to cover the study area across the surveys, and in total, 455 individuals were identified within the 36-month sampling period, with variable sampling effort, which was greater during the summer months (May–August). However, information was also available for all seasons of the year.

## Validation test

Validation aids in determining the degree to which a model is an accurate representation of the empirical wild populations from the model's intended perspective (*Trucano et al., 2006*). In this sense, the results obtained in the simulations under the different scenarios were compared with the empirical data of bottlenose dolphin populations from Alvarado, Mexico and Savannah, USA.

First, the individual sighting histories from the empirical data were categorized using the definitions of R and SF previously described (Tables 1 and 2). Simulations with similar resighting and survival values to the empirical data were then used for comparisons. For this, we used the apparent survival (*Phi*) (*i.e.,* the probability that any of the marked individuals survives and returns to the study area) values for each population as computed under the J-S approach in SOCPROG v2.9 software (*Whitehead, 2009*). It is noteworthy that both Savannah, USA, and Alvarado, Mexico dolphins showed the same survival (*Phi* = 0.92) which we used to compare with the simulated estimates of R and SF.

The available data on annual sighting probability for the Alvarado population by *Bolaños Jiménez et al. (2021)* was also used as a reference ($p = 0.88$); in this case, the high value refers to individuals having enough resightings such that they may be classified by any

of the definitions of R and SF applied in this study, thus it logically rules out transient individuals. The differences between the simulations and the empirical data were then assessed using a contingency table, using the Chi-square or the Mann–Whitney test where appropriate.

## Performance

Metrics to assess R and SF utilize different criteria for individual occurrence under different measurement scales (*i.e.,* monthly, seasonal, and annual). Therefore, we used the *Gower (1971)* distance to compare similarities among individual sighting histories. This measure determines how different two observations are in the presence of mixed data, that is, a combination of numerical and categorical information (*Pavoine et al., 2009*; *D'Urso & Massari, 2019*).

The definitions of R and SF considered the different variable types (*e.g.,* numerical and categorical) and measurement scales to calculate the Gower distance. As shown in Tables 1 and 2, the multi-categorical scale was evaluated as a "fuzzy" variable (*i.e.,* several columns were used to define the different possible levels of R and SF in each annual period, such that an individual could belong to different categories throughout the study).

Therefore, once the distance matrix was obtained, the contribution of the result of each index to the general Gower distance was calculated, based on the correlations between the distances obtained for each variable and the global distances obtained by mixing all the variables (*Bello et al., 2021*). In this sense, the contribution of each index to the dissimilarity was calculated using the "k.dist.cor" function of the "ade4" package (*Thioulouse et al., 2018*) in the R-project programming language (*R Core Team, 2024*). These contributions helped to identify indices that have the best resolution or structure for the assessment of residency and site fidelity to determine the similarity among individual sighting histories.

These indices were first applied to data from the two empirical populations (*i.e.,* Alvarado, Mexico and Savannah, USA) and then plotted with the simulated data previously described, for comparison to the simulated scenarios (*i.e.,* when resighting and apparent survival probabilities are both low and/or high) using a violin plot. These results provided the expected statistical distribution of the Gower distances for the selected indices of residency and site fidelity under all the simulated scenarios as a measure of their performance related to the observed (*i.e.,* empirical) values. All statistical analyses were conducted using R-project (*R Core Team, 2024*).

## RESULTS

### Calibration

The initial methodology of 10,000 simulations proved computationally intensive given our multiple combinations of parameters for the scenarios, thus the number of simulations was progressively reduced until empirical results showed that output values for the analyzed variables were stable; this occurred at approximately 100 runs, which was set as the optimal number for all the experiments. We also made trials using a range of superpopulation sizes (*i.e.,* $N = 100$-$600$) and noted that results did not alter the trends in the data, as these were expressed in proportions; however, confidence intervals narrowed as the number
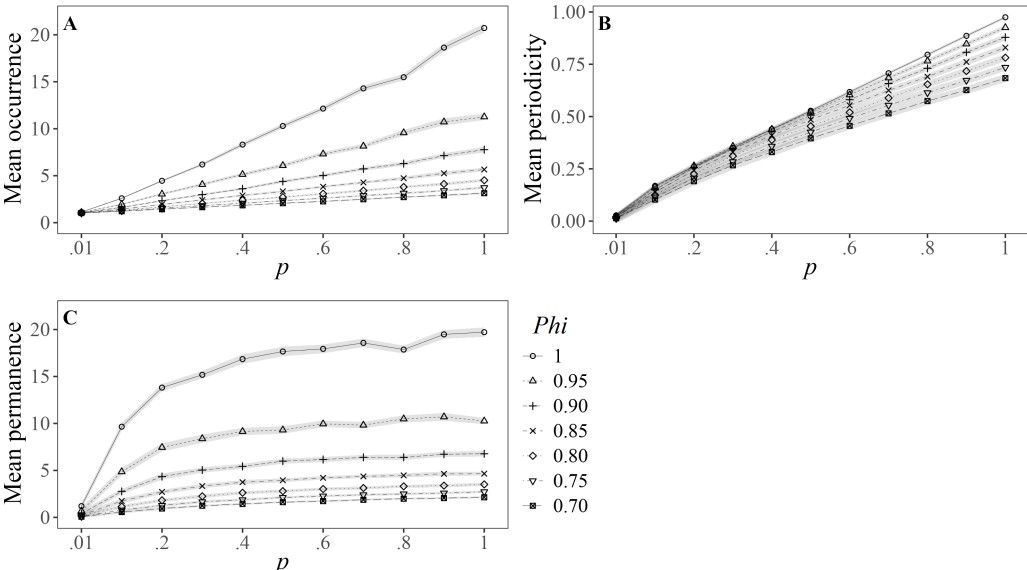

**Figure 1** **Mean values (±SD) for** *Ballance*'s (*1990*) **variables of residency: (A) Occurrence (total number of months sighted); (B) periodicity (inverse of the number of months between consecutive sightings), and (C) permanence (number of months between first and last sighting).** Results are based on 100 simulations for super-populations of individual bottlenose dolphins ($N = 600$), using monthly surveys during a three-year period under different probabilities of resighting ($p$) and apparent survival (*Phi*).

of simulated individuals available for sampling increased; thus, we used the largest N. As expected, the values for the parameters: superpopulation size ($N = 600$), entry probability ($b_t = 0.05$ to $0.15$), resighting probability ($p = 0.01$ to $1.00$), apparent survival probability (*Phi* = 0.70 to 1.00), sampling occasion ($t_i$, $I = 1$ to 36), sampling frequency ($m$ = once a month) and sampling duration (three years), all conditioned the number of marked individuals ($n$) available for sighting, and thus affected the computations of R and SF.

### Sensitivity

The results for Ballance's variables are represented in Fig. 1. The number of sighting records (*O sensu*; *Ballance, 1990*) showed a linear pattern under the different values of capturability ($p$), where the higher the survival (*Phi*), the greater the effect (slope) (Fig. 1A), and none reached the maximum value of 36 sightings (*i.e.,* present in all sampled months). In the case of the permanence, expressed as the number of months between the first and last sightings (*P sensu*; *Ballance, 1990*), all curves showed a different asymptote for the selected range of survival probabilities (Fig. 1B). The mean number of months between consecutive sightings ($I$ = inverse of the periodicity, *sensu*; *Morteo, Rocha-Olivares & Morteo, 2012*) had slight deviations in the slopes of all scenarios in the simulated data, which yielded in significant differences as the capture probability ($p$) increased (Fig. 1C).

Only eight of the evaluated indices classified individuals in some degree of R, and these were run in the simulations showing contrasting results. Classifications by *Rosel et al. (2011)*, *Dinis et al. (2016)*, and *Conway (2017)* only had two types (degrees) of R. For instance, estimates under *Rosel et al. (2011)* classification (Fig. 2A) showed a similar pattern

to the definition by *Conway (2017)* (Fig. 2C), managing to identify about 25% of resident individuals in unlikely favorable conditions (*e.g.*, $p = 1.00$, and *Phi* = 0.95). However, when survival and catchability are at their maximum, about 50% of the individuals may be classified as residents and non-residents. Also, under more probable conditions (*i.e.,* with intermediate values for resighting probabilities) it was possible to distinguish a reasonable proportion of resident individuals (*i.e.,* 25%), but only under high survival probabilities (>0.95).

The simulations under the classification by *Dinis et al. (2016)* showed that both in poor (*i.e.,* low probabilities of resighting and apparent survival), and in optimal conditions (*i.e.,* where *p* and *Phi* are close to 1.00), results were not very different, except for populations with maximum survival and capturability above 0.40 (Fig. 2B). Thus, this index is very strict in classifying resident individuals, managing to include at best 25% of them as residents under the most optimal scenarios.

Simulations for *Zolman (2002)*, showed that the proportion of individuals in the different degrees of R clearly vary under different values of survival. However, individuals were more likely to be classified as transient compared to the other categories despite the varying survival values; also, except for *Phi* = 1.00, individuals showed higher chances of being identified as residents than seasonal residents (Fig. 3A).

*Ananias, Jesus & Yamamoto*'s (*2008*) index was sensitive to temporality because for low values of *p* and *Phi*, it was more likely to classify individuals as temporary residents (Fig. 3B), whereas, with high probabilities of *p* and *Phi*, the probability of classifying annual resident individuals was higher. Temporary residents (*i.e.,* occasional visitors) were surprisingly scarce even with high survival values (*Phi* $\approx$ 1.00), compared to the other definitions.

Simulations under the *Martin & Silva (2004)* definition of R rarely classified individuals as residents (Fig. 3C). Only at the highest survival and resighting probabilities ( $\approx$1.00) was it possible to identify up to 25% of the individuals as residents.

On the other hand, the definition by *Möller, Allen & Harcourt (2002)* was more relaxed in identifying transient and resident individuals for both low and high survival and resighting values, respectively; conversely, individuals were less prone to be classified as occasional visitors (Fig. 3D) for resighting values over 0.10, and this effect was counterintuitively higher for the maximum survival value.

Finally, simulations under the definition by *Chabanne et al. (2012)* had a higher chance to detect transients and occasional visitors (*i.e.,* partial residents) (Fig. 3E) and were more conservative when classifying residents, as these occur only at $p > 0.50$ and *Phi* $\approx$ 1. Survival played an important role in the classification of occasional visitors since these were less likely to be identified at *Phi* > 0.95 and $p > 0.20$, turning them into resident individuals.

## Validation test

The application of the R definitions to simulated data showed that in most cases different proportions of individuals in each category occurred as compared to empirical datasets (Table 3); however, patterns were similar. For instance, *Martin & Silva (2004)* index failed to classify individuals as "permanent residents" in both simulated and empirical data; also, *Möller, Allen & Harcourt (2002)* definition showed higher chances to classify resident

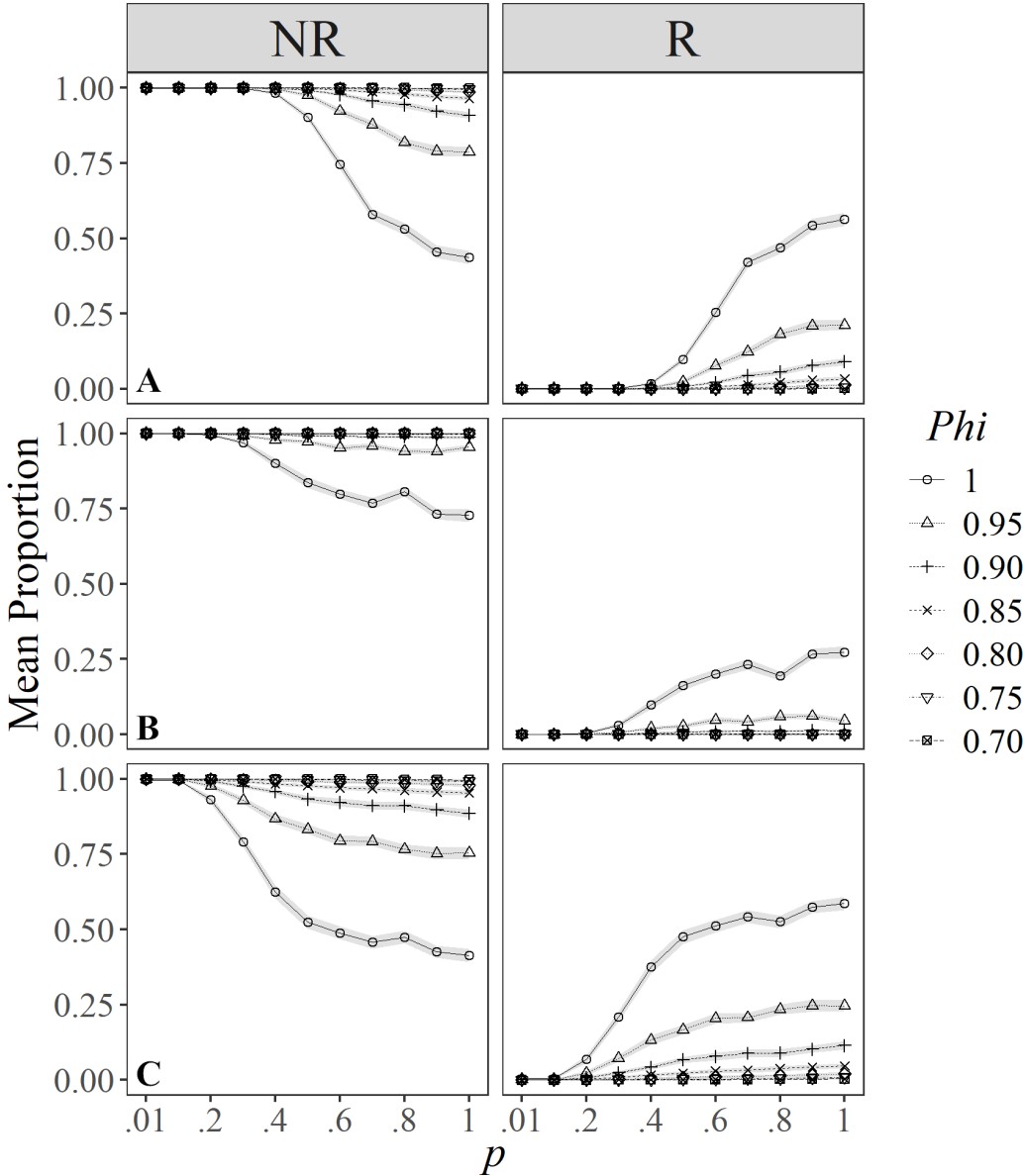

**Figure 2** Mean proportion (±SD) of individuals classified in residency categories (NR = Not resident, R = resident) by their sighting histories using the definitions by: (A) *Rosel et al. (2011)*; (B) *Dinis et al. (2016)*, and (C) *Conway (2017)*. Results are based on 100 simulations for super-populations of ($N = 600$) individual bottlenose dolphins, using monthly surveys during a three-year period under different probabilities of resighting ($p$) and survival (*Phi*).

individuals in both cases (simulated and empirical). The opposed trend was found in the case of *Rosel et al. (2011)* and *Dinis et al. (2016)*. Furthermore, the indices that included occasional visitors had similar patterns for simulations and empirical data, including *Ananias, Jesus & Yamamoto (2008)* and *Chabanne et al. (2012)*. Nevertheless, simulated data fitted the distribution for the Savannah dataset only under *Chabanne et al. (2012)*

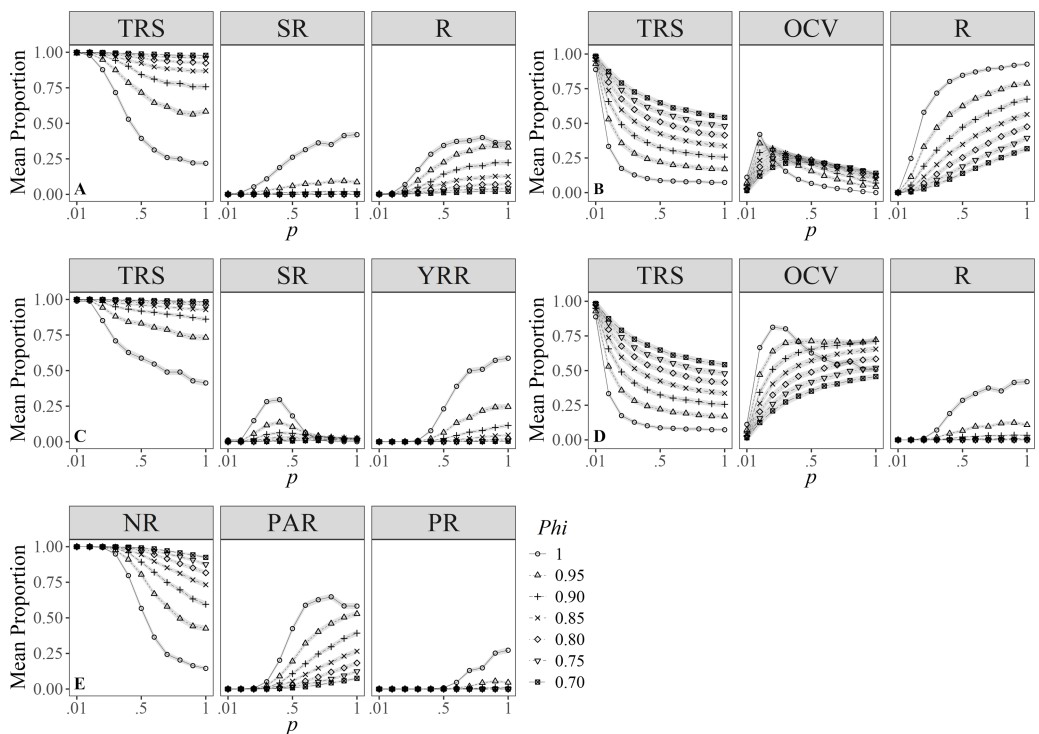

**Figure 3 Mean proportion (±SD) of individuals classified in residency categories.** (TRS, Transients; SR, Semi-residents; R, Residents; YRR, Year-round residents; OCV, Occasional visitors; NR, Not residents; PAR, Partial residents; PR, Permanent residents) by their sighting histories using the definitions by: (A) *Zolman (2002)*, (B) *Ananias, Jesus & Yamamoto (2008)*, (C) *Martin & Silva (2004)*, (D) *Möller, Allen & Harcourt (2002)*, (E) *Chabanne et al. (2012)*. Results are based on 100 simulations for super-populations of ( $N = 600$ ) individual bottlenose dolphins, using monthly surveys during a three-year period under different probabilities of resighting ($p$) and survival (*Phi*).

classification, as well as in *Dinis et al. (2016)* and *Conway (2017)*, and these last two also fitted the Alvarado dataset (Table 3).

On the other hand, empirical results (Alvarado, Mexico and Savannah, USA) showed similar values for the three Ballance's parameters, but the occurrence and periodicity were lower, whereas permanence was higher compared to simulations (Table 4). In this case, only the Periodicity was significantly different for both study cases.

As described earlier, from the 11 SF indices, seven were numerical, three were categorical and one had multiple categories. On the other hand, from the R indices, five were treated as categorical and three as "fuzzy" variables.

The contribution of each SF index to the total Gower distance for both empirical datasets showed a comparable pattern, where similarities among individuals were well represented in most cases (*i.e.*, contributions >0.70) (Fig. 4). The greatest contribution corresponded to the occurrence classification by *Balmer et al. (2008)*, the IH4 by *Tschopp et al. (2018)*, the monthly and seasonal sighting rates by *Chabanne et al. (2012)*, and *Ballance (1990)* (occurrence), whereas the least contribution corresponded to *Ballance (1990)* (periodicity parameter).

**Table 3  Proportion of individuals (±s.d.) classified in each category of residency under different definitions using the Jolly-Seber approach.** We used 100 simulations of super populations ($N = 600$ individuals) *vs.* empirical data from wild dolphin populations surveyed monthly across a three-year period (Common bottlenose dolphins $N_{Alvarado} = 206$ individuals; Tamanend's bottlenose dolphins $N_{Savannah} = 455$).

| Index | Category | Simulated data | Empirical data | |
|---|---|---|---|---|
| | | (%) | Alvarado, Mexico (%) | Savannah, USA (%) |
| *Ananias, Jesus & Yamamoto (2008)*[a,b] | TRS | $64 \pm 1.02$ | 80 | 70 |
| | RS | $14 \pm 0.72$ | 17 | 27 |
| | RY | $22 \pm 1.04$ | 3 | 3 |
| *Chabanne et al. (2012)*[a] | TRS | $23 \pm 1.64$ | 40 | 18 |
| | OCV | $72 \pm 1.76$ | 52 | 76 |
| | R | $6 \pm 1.03$ | 8 | 6 |
| *Conway (2017)* | NR | $77 \pm 1.02$ | 87 | 82 |
| | R | $23 \pm 1.07$ | 13 | 18 |
| *Dinis et al. (2016)* | NR | $98 \pm 0.56$ | 100 | 97 |
| | R | $2 \pm 0.56$ | 0 | 3 |
| *Martin & Silva (2004)*[a,b] | NR | $58 \pm 1.89$ | 95 | 96 |
| | PR | $40 \pm 1.87$ | 5 | 4 |
| | RP | $0.00 \pm 0.00$ | 0 | 0 |
| *Möller, Allen & Harcourt (2002)*[a,b] | TRS | $23 \pm 1.64$ | 40 | 18 |
| | OCV | $8 \pm 1.12$ | 17 | 25 |
| | R | $69 \pm 1.88$ | 42 | 57 |
| *Rosel et al. (2011)*[a,b] | NR | $82 \pm 0.93$ | 98 | 99 |
| | R | $18 \pm 0.93$ | 2 | 1 |
| *Zolman (2002)*[a,b] | TRS | $70 \pm 1.85$ | 86 | 83 |
| | RS | $4 \pm 0.86$ | 3 | 7 |
| | R | $26 \pm 1.77$ | 11 | 10 |

**Notes.**

TRS, Transient; RS, Seasonal resident; RY, Yearly residents; OCV, Occasional visitors; R, Resident; NR, Not resident; PR, Partial resident.

Significant differences (Chi-square, $p < 0.05$) between simulations and empirical are shown for Alvarado (a) and Savannah (b).

Conversely, the contribution of each R index to the total Gower distance for both empirical datasets were highly variable. For instance, computations for the Alvarado dataset were not possible under the *Dinis et al. (2016)* definition, since all individuals were classified as not resident. Moreover, contributions were well represented (*i.e.,* contributions >0.70) for the Alvarado dataset under four of the definitions (*i.e., Zolman, 2002*; *Ananias, Jesus & Yamamoto, 2008*; *Chabanne et al., 2012*; *Conway, 2017*), but only in one instance (*i.e., Chabanne et al., 2012*) for the Savannah population (Fig. 5).

## Performance

SF indices were not tested for performance with the simulated data, given the high redundancy in the information; that is, high co-variability was previously observed among all the indices. However, the simulations of sighting histories with the apparent survival

**Table 4** Mean (±s.d.) estimates of *Ballance*'s (*1990*) parameters using the Jolly-Seber approach.

| Parameter | Simulations ($N = 600$) | Empirical data | |
|---|---|---|---|
| | | Alvarado, Mexico ($N = 206$) | Savannah, USA ($N = 455$) |
| Occurrence | 8.02 ± 0.27 | 4.27 ± 4.10 | 5.43 ± 3.82 |
| Permanence | 7.86 ± 0.30 | 12.64 ± 12.90 | 18.46 ± 12.24 |
| Periodicity | 0.80 ± 0.01 | 0.19 ± 0.24[*] | 0.24 ± 0.22[*] |

Notes.

We used 100 simulations of super populations ($N = 600$ individuals) *vs.* empirical data from wild bottlenose dolphin populations (Common bottlenose dolphins $N_{Alvarado} = 206$ individuals; Tamanend's bottlenose dolphins $N_{Savannah} = 455$) surveyed monthly across a three-year period.

*Significant differences between empirical and simulated data (Mann-Whitney, $a = 0.05$).

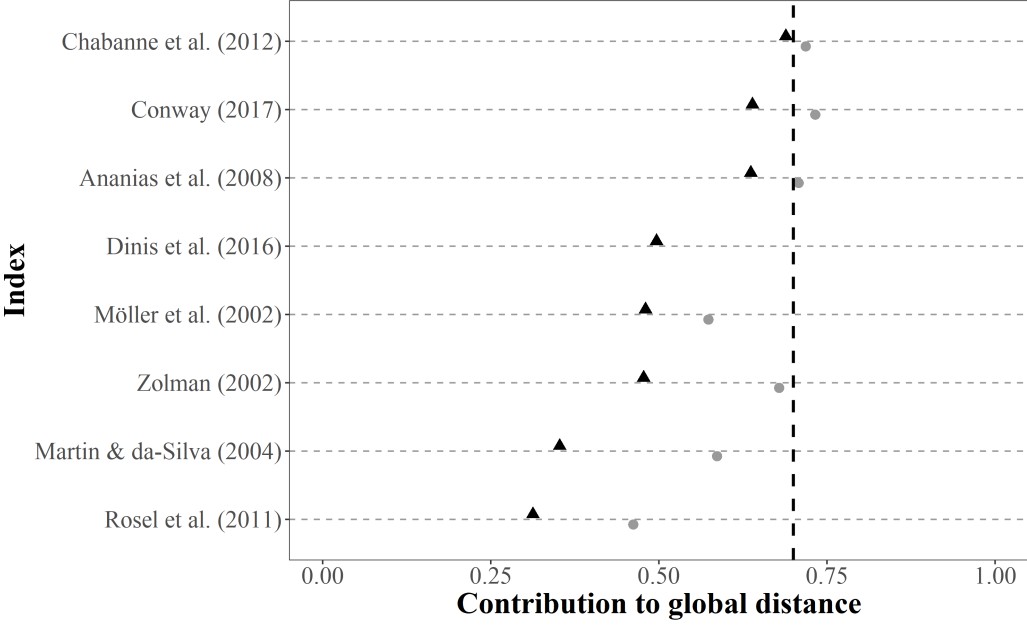

**Figure 4** Dot plot of the contribution (similarities among the classification of sighting histories from individuals) of different SF indices to the global Gower distance. We used two empirical bottlenose dolphin populations surveyed monthly across a three-year period (Alvarado, Mexico $N = 206$ individuals; Savannah, USA $N = 455$ individuals). Notations from the descriptions in the original papers were used for distinction (*i.e.,* IH4, SSR, MSR, SR and YSR), and the Values to the right of the dotted line (0.70 contribution to global distance) had the highest contributions.

probability of the empirical data (*i.e.,* 0.92), showed that the R indices with the highest contributions were those of *Ananias, Jesus & Yamamoto (2008)*, *Chabanne et al. (2012)*, *Conway (2017)*, and *Möller, Allen & Harcourt (2002)* for different resighting rates (Fig. 5), although these were highly variable. The contribution for most of the indices was within the simulated intermediate values, but the simulations showed that the classification by
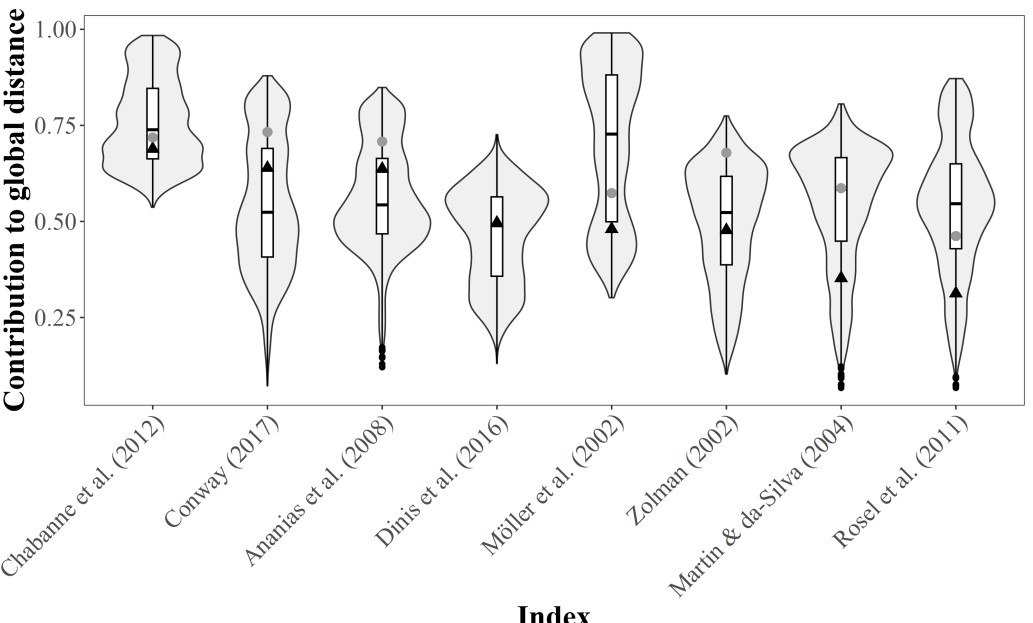

**Figure 5** **Violin plot for the contribution differences of residency (R) indices to the global Gower distance.** We used simulated data on bottlenose dolphin mark-resight histories (grey area, $N = 100$ simulations) under the Jolly-Saber framework, using the same survival probability of the empirical datasets (*Phi* $= 0.92$) and all possible resighting scenarios ($0.01 \geq p \leq 1.00$). Medians (horizontal lines), 25 –75% quartiles (boxes), and value ranges (whiskers) are also shown. The dot and triangle represent point estimates for the indices from the two empirical bottlenose dolphin populations (Alvarado, Mexico $N = 206$ individuals and Savannah, USA $N = 455$ individuals, respectively).

*Chabanne et al. (2012)* had more consistent results, due to less variability in the classification of the simulated resighting scenarios. Furthermore, the empirical data fell within the intermediate values of the simulated data (Fig. 5).

## DISCUSSION

As expected, simulations for R and SF showed sensitivity under different values of survival and resighting (*Morteo, Rocha-Olivares & Morteo, 2012*), although the magnitude and direction of the changes were previously unknown. The R variables by *Ballance (1990)* showed a positive trend in relation to both parameters. On the other hand, the different measures of SF showed similar increasing trends, which highlight the importance of the relationship between the probability of apparent survival and the probability of resighting (*i.e.,* the probability of resighting an animal depends on it being alive) (*Hammond, 2009*).

Although all the indices in this study were classified and evaluated independently, in practice, many of these assess R and SF jointly or indistinctly (*e.g., Möller, Allen & Harcourt, 2002*; *Chabanne et al., 2012*; *Huesca-Domínguez et al., 2024*). However, we emphasize that SF indices provide a way to summarize the frequency of sightings for an individual in a study area regardless of the temporal scale (seasonal or annual) and sampling resolution,
and thus are better suited for inter-study comparisons. Although we would suggest using those with the greatest contribution to Gower's global distance, their selection will strongly depend on the case under study (*Huesca-Domínguez et al., 2024*).

Based on both our empirical datasets (Alvarado, Mexico and Savannah, USA), the best methods to represent SF were those by *Balmer et al. (2008)*, *Tschopp et al. (2018)*, but also the occurrence by *Ballance (1990)* and the monthly and seasonal sighting rates of *Chabanne et al. (2012)*, since these scored the highest in performance and showed close values for the data of our empirical cases during validation (Fig. 5).

It is noteworthy that the IH4 (*Tschopp et al., 2018*) grouped different indices of SF into one, and standardized occurrence and permanence (*sensu Ballance, 1990*) on a scale from 0 to 1 to generate an index, which showed better performance compared to the separate use of the variables by *Ballance (1990)*. On the other hand, *Balmer et al. (2008)* classified the frequency of occurrence on a categorical scale preventing information loss on the numerical occurrence when it is transformed into a categorical variable. Thus, if a categorical resolution of the occurrence is desired, the definition by *Balmer et al. (2008)* will be more suitable. However, the number of categories can vary between studies, which can prevent comparisons, but it can serve as a descriptive measure of the shape of the distribution of the occurrence if it is plotted in a histogram (*Balmer et al., 2008*; *Morteo, Rocha-Olivares & Morteo, 2012*).

Conversely, our results for the R indices were highly variable and contrasting depending on the definitions and criteria used by the different authors. Furthermore, definitions with three R categories were more heterogeneous than those with two categories, where the main discrepancies occurred in the intermediate category; that is, when considering occasional individuals or seasonal residents. In these cases, the main differences among categories were based on the definitions and thresholds; thus, in some cases, seasonal residents may be equivalent to occasional visitors. The latter yields counterintuitive proportions of individuals classified in some of the intermediate categories in relation to "unusually high" values of apparent survival and catchability.

As expected, individuals with high levels of survival and thus resightings are classified as residents more frequently, since they are easier to identify than migratory individuals (which have close to zero apparent survival and therefore are rarely observed in the sampling period; *Pradel et al., 1997*). This confirms the modulatory role of resighting values as identified by *Morteo, Rocha-Olivares & Morteo (2012)*, which are dependent on the sampling frequency and adds apparent survival as a factor in the classification of individuals in the different R categories.

It is important to note that even when simulations did not explicitly consider differences in population structure derived from sex and age in bottlenose dolphins, these can be inferred from the modeled heterogeneity in survival values and sighting rates, leading to the expected behavior of males and females, as well as individuals in different developmental stages. This is becoming relevant, as variables such as sex, age, size of individuals, and even social bonds have been incorporated recently in demographic studies on dolphins for their potential to create heterogeneity in capture or apparent survival probabilities (*Bolaños Jiménez et al., 2021*). Our approach is justified in different studies that report

higher resighting and survival in females, and lower values for males and calves (*Currey et al., 2009*; *Arso Civil et al., 2019*; *Bolaños Jiménez et al., 2021*; *Ludwig et al., 2021*). In this sense, simulated scenarios were planned based on empirical data (Alvarado, Mexico and Savannah, USA), and then calibrated such that simulations include the differences that may be found in structured populations of both studied species (*i.e., T. truncatus* and *T. erebenus*, respectively). Admittedly, the sex and age of these animals are commonly unknown (*Bolaños Jiménez et al., 2021*), and the parameters used to define R and SF are often exposed to different sources of variation derived from field logistics, as well as from the local geography and ecology of the study areas, and these can have unpredictable repercussions in the results of each study (*Balmer et al., 2014*; *Dinis et al., 2016*).

The Gower's global distance showed evidence of the efficiency of the R and SF indices in each case study (Alvarado, Mexico and Savannah, USA) since it uses quantitative, categorical, and "fuzzy" data to assess the temporal patterns of individuals in a given period. The contribution to the global Gower's distance determines how capable each index was of producing similar values (or not) for individuals, based on their temporal patterns, using a set of metrics determined for the population under study. Thus, the global distance obtained for the two case studies showed that the indices by *Chabanne et al. (2012)*, *Ananias, Jesus & Yamamoto (2008)*, and *Conway (2017)* had a better contribution compared to other definitions. These three indices highlight the importance of the temporal scale in R evaluations, providing a better resolution to classify individuals as seasonal or year-round residents; thus, these are the most recommended indices to define R.

Admittedly, as pointed out by *Huesca-Domínguez et al. (2024)* using a temporal scale is challenging when carrying out R assessments, since not all studies use the same sampling duration, number of sampling events, and definitions of seasons. Therefore, the use of different temporal scales should be evaluated to determine which metric would be more consistent or would fit best the dynamics of the population under evaluation for the assessment of R. Note that the use and efficiency of these will depend in each case both on the cutting points of the definitions to classify individuals and on the ability of researchers to carry out exhaustive sampling surveys with the resolution and duration to resolve individuals under the effects of short term or seasonal variations (*Srinivasan, 2018*; *Huesca-Domínguez et al., 2024*). For instance, given the comparative nature of this research, both our case studies were evaluated using the same temporal scale (four seasons per year). However, this does not necessarily apply specifically to the Alvarado, Mexico population, whose habitat is commonly characterized by only three seasons (*Morteo, Rocha-Olivares & Morteo, 2012*; *Morteo, Rocha-Olivares & Abarca-Arenas, 2017*; *Bolaños Jiménez et al., 2021*), a situation that remains to be assessed. Still, the use of this approach allowed us to observe that, although the contribution of each metric was similar in both cases, the method by (*Möller, Allen & Harcourt, 2002*) had better results when dealing with data gaps across spatial and temporal scales. Thus, it may be advisable as an alternative to the methods by *Ananias, Jesus & Yamamoto (2008)*, *Chabanne et al. (2012)*, and *Conway (2017)*.

Our results contribute to the ongoing discussion on methodological implications for the interpretation of ecological patterns in bottlenose dolphins. As migratory movements of individuals are known to occur in a wide range of temporal and spatial scales, scientists

and conservation agencies require more detailed data for making informed decisions for managing stocks and populations worldwide. In that sense, spatial overlap within different timeframes (*e.g.*, seasons or years) as well as sudden or gradual changes in distribution will challenge the ability for any R and SF methodology to assess and detect resident and transient individuals, thus introducing biases and increasing uncertainty (*Haughey et al., 2020*; *Durden et al., 2021*). This is especially relevant when modeling demographic parameters and may also lead to confusion when interpreting home ranges and habitat use on a population level (*Haughey et al., 2020*; *Ludwig et al., 2021*). Therefore, accounting for different types of residency or degrees of transience across the duration of the studies is important to understand variations in population structure, social dynamics, gene flow, and behavioral adaptations (*Conn et al., 2011*; *Bolaños Jiménez et al., 2021*; *De Moura et al., 2021*; *Durden et al., 2021*; *Nicholson et al., 2021*; *Pace et al., 2021*; *Hohn et al., 2022*).

In summary, R and SF assessments must be addressed within the constraints and the magnitude of the biases inherent for each method and study area (*Rossi-Santos, Wedekin & Monteiro-Filho, 2007*; *Tschopp et al., 2018*). That is, acknowledging the sensitivity (capturing true positives) and specificity (avoiding false positives) of each R and SF index provided in this paper, along with the limitations of the data, would help researchers obtain consistent results that are more attuned to the reality of each bottlenose dolphin population and thus favor the chances of controlling or at least identifying biases. The latter would promote better management policies in favor of better conservation efforts and will also allow a more standard approach for comparisons among studies (*Morteo, Rocha-Olivares & Morteo, 2012*).

It is worth noting that the methods for assessing R and SF evaluated here, are based on subjective criteria developed to meet the specific needs of research studies mostly within the genus *Tursiops*. Therefore, these results could be tested with more objective methods such as hierarchical cluster analysis as used in several recent studies (*e.g.*, *Zanardo, Parra & Möller, 2016*; *Haughey et al., 2020*; *Cipriano et al., 2022*; *Mintzer, Quackenbush & Fazioli, 2023*), including using the Gower distance for mixed data due to the flexible nature of the multiple indicator scales (*e.g.*, *Pace et al., 2021*). Also, the use of metrics developed for other species will help to gain insight in terms of their similarities, potential applicability, and would aid in fine tuning their criteria and thresholds to achieve higher consistency and lower biases.

Finally, considering that residency is a latent and unobservable process that cannot be inferred solely from sighting records (*Conn et al., 2011*), future research could improve the experimental design by including other common sampling strategies (*e.g.*, based in survey effort, photographic or identification success rates, spatial densities), other types of data (*e.g.*, genetic, acoustic, satellite tags), and more complex parametrizations for population models. Technical advances could also provide automated assessments of R and SF while expanding classifications to other definitions and criteria, and even contrasting the results to the expected values given the parameters for each population and survey design.

## ACKNOWLEDGEMENTS

This work is part of the first author's MSc thesis in Integrative Biology at Universidad Veracruzana. The authors wish to acknowledge the contribution of Jessica Powell for providing valuable insight into management applications of this research and linking together many of the authors for this collaborative project. In addition, we are thankful for Reagan Munday for providing bibliography on the topic. We would also like to acknowledge Robin M. Perrtree for her dedication and for leading all the data collection and management for the Savannah, USA case study. We would also like to acknowledge all the staff and volunteers that participated in the collection of the empirical data from both the Alvarado, Mexico and Savannah, USA study sites, without their hard work in the field and lab, this project would not have been possible.

### Funding

This work was supported by Penn Clarke and Dolphin Relief & Research. The funders had no role in study design, data collection and analysis, decision to publish, or preparation of the manuscript.

### Grant Disclosures

The following grant information was disclosed by the authors:
Penn Clarke and Dolphin Relief & Research.

### Competing Interests

The authors declare there are no competing interests.

### Author Contributions

- Israel Huesca-Domínguez conceived and designed the experiments, performed the experiments, analyzed the data, prepared figures and/or tables, authored or reviewed drafts of the article, and approved the final draft.
- Eduardo Morteo conceived and designed the experiments, performed the experiments, analyzed the data, prepared figures and/or tables, authored or reviewed drafts of the article, and approved the final draft.
- Luis Gerardo Abarca-Arenas conceived and designed the experiments, performed the experiments, analyzed the data, authored or reviewed drafts of the article, and approved the final draft.
- Brian C. Balmer conceived and designed the experiments, authored or reviewed drafts of the article, and approved the final draft.
- Tara M. Cox conceived and designed the experiments, authored or reviewed drafts of the article, and approved the final draft.
- Christian A. Delfín-Alfonso conceived and designed the experiments, performed the experiments, authored or reviewed drafts of the article, and approved the final draft.
- Isabel C. Hernández-Candelario performed the experiments, authored or reviewed drafts of the article, and approved the final draft.

## Data Availability

The data and scripts are available at GitHub and Zenodo:

– https://github.com/ihuesca/R_SF_simulations.

– Huesca, I. (2024). Method selection affects the estimates of residency and site fidelity in bottlenose dolphins: Testing sensitivity and performance of different methods using mark-resight data [Data set]. Zenodo. https://doi.org/10.5281/zenodo.13381701.

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
