# Peer review of "Method selection affects the estimates of residency and site fidelity in bottlenose dolphins: testing sensitivity and performance of different methods using mark-resight data"

_PeerJ, doi:10.7717/peerj.18329_

## Round 0.1 · original submission · Major Revisions

Both reviewer agreed that this is an interesting and well-written article. However there were some issues that needs to be addressed referring to reviewer 2's comment on evaluation of the residency and site-fidelity indices and also updating the literature to more recent ones.

Please check all format and also improvements recommended

·

Basic reporting

The article is very well written. I have just one comment on the formatting: why is there a line break between paragraphs in the Introduction and not in the Material and Methods and Results sections? The line break reappears during the discussion.

The references are numerous. Attention, line 520: you quote Bolanos et al. 2021... It must be Bolanos-Jimenez et al. 2021 (already cited before), otherwise this reference is not in the list.

Experimental design

This is an analytical article that challenges various models and hypotheses.

It may not be original in its subject matter, but it is a very useful article. Useful for all those who use the notions of residency and site fidelity in their studies of marine mammal populations.

The presentation of the problem is well done, and it's easy to see why these principles should be studied in greater detail.

The materials and methods section is well described so that all results can be followed.

Validity of the findings

While the results are well presented, I find the discussion too long and, above all, it's based on direct discussion of the analysis results without really putting them into perspective with precisely the interest of this study for those who use these models.

We could suggest that the authors synthesize their discussion, which resembles a dissertation discussion, and orient it more towards the direct use suggested by their results.

The end of the article would gain in clarity, and would then open up its field of interest not only to specialists in model analysis, but to all ecologists.

Reviewer 2 ·

Basic reporting

The manuscript is well-written with good English and has a great flow. It was a pleasure to read and is within the scope of PeerJ. This manuscript would be of great interest to readers, in particular population biology researchers and managers of dolphin populations globally. Saying this, I think there is some work that the authors need to do to assure the reader that the information being shared is current.

If this paper was published in 2020, I would not have concerns about the literature the study is based on. However, given it is mid-2024, a lot more literature is available on this topic. More current literature should be incorporated throughout the manuscript, including in the review for site-fidelity and residency metrics.

I am concerned that the review on how site fidelity and residency patterns are assessed is not necessarily based on current literature, as reviewing thesis by Huesca-Dominguez shows that this review included scientific literature between 1990 and 2019. A quick Google scholar search on site fidelity bring up 30+ relevant publications post 2020. I think it is necessary that the authors review site fidelity and residency metrics used to date and incorporate any new approaches in this study. If no new approaches are found, authors should justify not using more current literature by appropriately referencing to the newer literature, highlighting no difference in methods used.

No raw data was provided, but I think it is not applicable for this article. The authors could share the code they have used though.

All formatting needs to be double checked – line spacing was inconsistent in places.

The article is professionally structured with figures and tables, although captions to these need some work to be stand-alone.

Specific comments:

Title: This is a personal preference, but I would not have a question in the title but rather answer the question in the title. It also looks visually clunky with the ?: next to each other. I would go with something along the lines of ‘Choice of method affects the estimate of residency and site fidelity of bottlenose dolphins: Testing sensitivity and performance of different methods using capture-recapture data’


Abstract

Line 20: The title uses the term capture-recapture data, while here you use mark-resight data. Either use one term or define the difference between the two terms. This applies to all of the manuscript.

Line 26: Heterogeneity in what? I assume heterogeneity in site-fidelity?

Line 29: You make a great conclusion here, but I don’t see this same conclusion explicitly made in the discussion.

Line 39: Are you able to cut down in words so you can explicitly say what the residency definition my Möller et al. 2002 was. Also, your spelling of ‘Möller’ is incorrect here. Make sure you check reference spelling throughout the manuscript.

Introduction

Your introduction reads well and easy to understand. It is, however, noticeable that all references are from before 2021. Quite a few papers have been published on dolphin site fidelity and residency patters in the last few years. The introduction should include more recent work to show that this study is based on currently used methods in assessing site fidelity and residency patterns of dolphins.

Experimental design

Generally, the methods are written really well and are easy to understand. Research questions were well defined, relevant and meaningful with the research being a valuable contribution to primary literature. Methods are described in sufficient detail, although as per previous comments the excluding current studies, it may be necessary to re-evaluate if current methods of assessing residency and site-fidelity were incorporated in this study.

See specific comments below:

Line 106-111: Do I understand correctly that the work if Huesca-Dominguez et al. is the review that you refer to? It would be good to have a few sentences in summarizing the literature review methods. If the ‘in-press’ article will be published prior to the publication of this manuscript, then the reference may suffice.

Line 126: Here you go back to referring to capture-recapture models – define and use terms consistently to not confuse the reader. Check across the entire manuscript.

Line 129-131: You need supporting references across taxa here.

Line 134-135: Define what is meant by ‘superpopulation’.

Line 164: How did you choose the size of the superpopulation as 150 individuals? Depending on your definition (see above comment) I think this is a very low number for Tursiops spp. Justify the choice, perhaps even try a few options.

Line 259: Given fuzzy is not a common term (at least to what I am aware of) best to say ‘fuzzy’ – unless I am mistaken about the term’s frequency of use.

Line 194-195: This is where I feel it would be good to have confidence that the scenarios are really based on current methods. See my general comments on current literature not being incorporated.

Line 238-240: Would this not be apparent survival? Define the term and explain how this is calculated in SOCPROG v.2.9.

Tables and figures

Make sure all your figure and table captions are ‘stand-alone’ and include the species you are referring to.

Table 1 and 2. These are all for bottlenose dolphins, correct? Table caption should mention species.

Figure 4: Give notations as to what all the abbreviations on the y-axis are and why some of the references do not have any abbreviations.

Validity of the findings

No data or code was provided - model parameters were provided so one could replicate the study. I recommend the code used to be made available.

Again, the discussion lacks reference to published literature since 2021. Given multiple papers have been published since 2021 estimating residency and site-fidelity of dolphins, these should have been incorporated in the indices being evaluated, if they were different to indices used in studies referred to in this manuscript.

Results

Line 331-332: Make sure this is discussed more explicitly – especially keeping in mind the definition of survival and how it is estimated.

Line 375: For ease of following, here would be good remind the reader what the definition is together with the reference.

Line 402: Define to the reader in text what IH4 is.

Discussion

There are no references beyond 2021 despite publications having come out since 2020 on this subject. This does not mean that the arguments are not valid but would like to see some evidence of current literature having been incorporated.

Line 448: As mentioned before, be consistent with terms – resighting vs recapturing.

Line 499: Very important point. I think you could elaborate here a bit more to make the reader really understand what the recommended action here is. E.g., do you recommend managers look at the how R and SF were defined to determine how they may fraction populations for management. Sometimes, for example, an area may be important for transient individuals or seasonal residents (e.g., for feeding or mating – see for example Nicholson et al. 2023 DOI: 10.1002/aqc.3736) even if they don’t use the area all the time.

Line 513 -514: What is the expected behavior of males and females? Need to mention this and reference appropriately.

Line 605-606: I think the authors should include information on how the data was collected during all the studies whose indices have been used here – this could be reported in Table 1 and 2. A major consideration that may affect assessment of residency/site fidelity is whether the study was carried out temporally and spatially in a way to give each individual present in the study area during a sampling period an equal probability of being in the sample.

---

## Round 0.2 · accepted · Accept

Reviewers and myself are satisfied with the corrections done. For this I accept the manuscript to be published

·

Basic reporting

After verification, all the comments (both mine and those of the second reviewer) were taken into account.

I have no further comments.

Experimental design

After verification, all the comments (both mine and those of the second reviewer) were taken into account.

I have no further comments.

Validity of the findings

After verification, all the comments (both mine and those of the second reviewer) were taken into account.

I have no further comments.

Additional comments

After verification, all the comments (both mine and those of the second reviewer) were taken into account.

I have no further comments.